# Early Neonatal Meconium Does Not Have a Demonstrable Microbiota Determined through Use of Robust Negative Controls with *cpn*60-Based Microbiome Profiling

Scott J. Dos Santos,[a] Zahra Pakzad,[b,c] Chelsea N. Elwood,[b,d] Arianne Y. K. Albert,[b] Soren Gantt,[e] Amee R. Manges,[f] Tim J. Dumonceaux,[g] Evelyn J. Maan,[b] Janet E. Hill,[a] Deborah M. Money,[b,d] The Maternal Microbiome Legacy Project Team

aDepartment of Veterinary Microbiology, Western College of Veterinary Medicine, University of Saskatchewan, Saskatoon, Saskatchewan, Canada
bWomen's Health Research Institute, B.C. Women's Hospital, Vancouver, British Columbia, Canada
cDepartment of Microbiology and Immunology, Faculty of Science, University of British Columbia, Vancouver, British Columbia, Canada
dDepartment of Obstetrics and Gynaecology, Faculty of Medicine, University of British Columbia, Vancouver, British Columbia, Canada
eCentre de Recherche du CHU Sainte-Justine, Quebec, Canada
fSchool of Population and Public Health, Faculty of Medicine, University of British Columbia, British Columbia, Canada
gAgriculture and Agri-Food Canada, Saskatoon, Saskatchewan, Canada

**ABSTRACT** Detection of bacterial DNA within meconium is often cited as evidence supporting *in utero* colonization. However, many studies fail to adequately control for contamination. We aimed to define the microbial content of meconium under properly controlled conditions. DNA was extracted from 141 meconium samples and subjected to *cpn*60-based microbiome profiling, with controls to assess contamination throughout. Total bacterial loads of neonatal meconium, infant stool, and controls were compared by 16S rRNA quantitative PCR (qPCR). Viable bacteria within meconium were cultured, and isolate clonality was assessed by pulsed-field gel electrophoresis (PFGE). Meconium samples did not differ significantly from controls with respect to read numbers or taxonomic composition. Twenty (14%) outliers with markedly higher read numbers were collected significantly later after birth and appeared more like transitional stool than meconium. Total bacterial loads were significantly higher in stool than in meconium, which did not differ from that of sequencing controls, and correlated well with read numbers. Cultured isolates were most frequently identified as *Staphylococcus epidermidis*, *Enterococcus faecalis*, or *Escherichia coli*, with PFGE indicating high intraspecies diversity. Our findings highlight the importance of robust controls in studies of low microbial biomass samples and argue against meaningful bacterial colonization *in utero*. Given that meconium microbiome profiles could not be distinguished from sequencing controls, and that viable bacteria within meconium appeared uncommon and largely consistent with postnatal skin colonization, there does not appear to be a meconium microbiota.

**IMPORTANCE** Much like the recent placental microbiome controversy, studies of neonatal meconium reporting bacterial communities within the fetal and neonatal gut imply that microbial colonization begins prior to birth. However, recent work has shown that placental microbiomes almost exclusively represent contamination from lab reagents and the environment. Here, we demonstrate that prior studies of neonatal meconium are impacted by the same issue, showing that the microbial content of meconium does not differ from negative controls that have never contained any biological material. Our culture findings similarly supported this notion and largely comprised bacteria normally associated with healthy skin. Overall, our work adds to the growing body of evidence against the *in utero* colonization hypothesis.

**KEYWORDS** contamination, *cpn*60, low biomass, meconium, microbiome, negative controls, sequencing

Address correspondence to Janet E. Hill, janet.hill@usask.ca, or Deborah M. Money, deborah.money@ubc.ca.

**D**ysbiotic shifts in the human gut microbiome composition have been associated with clinically important chronic conditions, including asthma (1), type I diabetes mellitus (2), inflammatory bowel disease (3), obesity (4), and various autoimmune disorders (5). Supporting data from murine models demonstrate the role of a "healthy" microbiome in immune system development from birth, with germfree mice exhibiting aberrant immunologic phenotypes which persist into adulthood (6).

Accordingly, acquisition and development of the gut microbiome immediately following birth is emerging as a major focus of research (7, 8). Convention held that the fetus develops within a sterile environment, owing to the inability to culture microorganisms from amniotic fluid and placental tissue (9, 10). Recent studies have reported detection of microbial DNA in placental tissue (11, 12), amniotic fluid (12–14), and meconium (13, 15–19), initiating the *in utero* colonization hypothesis. However, many of these studies failed to account for routine sources of contaminating DNA (20–22). Prior to laboratory analyses, specimen collection and handling present significant opportunities for potential contamination. Common issues in the laboratory include failing to report sequencing data for negative controls, relying on gel electrophoresis to demonstrate lack of contamination, or even failing to include any controls (11–15, 17, 19, 23, 24).

Contamination of commercially available nucleic acid extraction kits and PCR reagents with bacterial DNA is well recognized (20, 21). Additional contaminants may be inadvertently introduced from the environment during library preparation and through sample cross-contamination (25). These contaminants are usually negligible when investigating body sites or environmental niches where bacteria are abundant; however, the issue becomes critical when studying biological material that is inherently low in microbial biomass, such as meconium or placental tissue. Here, contamination may account for a large proportion of the sequence data generated (25–28). Controlling for contamination is therefore essential when characterizing the microbiome of low-microbial biomass environments.

The existence of microbial communities within the placenta or fetal gut has major implications for the establishment and maturation of the neonatal gut microbiome. However, mounting evidence points to contamination as an overwhelming confounder in these studies. In order to address this issue, we employed *cpn*60-based amplicon sequencing to define the microbial composition of meconium obtained within 72 h of birth. Specifically, we sought to compare meconium microbiome profiles to those of negative controls, including DNA extraction kit reagents, extraction blanks, and mock samples. As a secondary objective, we estimated the total bacterial load of meconium and sequencing negative controls compared to stool collected from 3-month-old infants. Finally, we screened meconium samples for viable organisms using bacterial culture and assessed the clonality of the most frequently identified species to assess the likelihood that these isolates originated from a common, contaminating source.

## RESULTS

**Cohort description.** We studied a subset of 141 infants enrolled in the Maternal Microbiome Legacy Project, recruited in Vancouver, Canada. All 141 infants were delivered at term (Table 1; mean gestational age, 39.6 weeks $\pm$1.2); 57 infants (40.4%) were delivered vaginally, while 84 (59.8%) were delivered by caesarean section (56 elective, 28 emergent). The proportion of caesarean section-delivered infants was per study protocol and does not reflect the institution's caesarean section rate of approximately 32.6% (29). As per institutional policy, all 86 women delivering by caesarean section received antibiotics at least 1 h prior to delivery. Another 11 women delivering vaginally also received antibiotics prior to delivery, indicated by GBS (group B *Streptococcus*) status or a diagnosis of chorioamnionitis. Eight women exhibited fever during labor and delivery, three of which were diagnosed with chorioamnionitis based on clinical findings. Meconium-stained amniotic fluid was observed in 22/141 infants (14.2%) during labor and delivery.

**Microbiome profiling.** Prior to sequencing, we confirmed the presence of amplifiable DNA in meconium through PCR of the human mitochondrial COX-1 gene (see Fig. S1 in the supplemental material), ruling out PCR inhibition as a potential confounder. Following

**TABLE 1** Demographics of the study population (141 neonates from 138 mothers)

| Characteristic | Delivery mode[a] | | |
| --- | --- | --- | --- |
| | Vaginal | C/S elective | C/S emergent |
| Mother's age (yrs) (n = 138) | 35.4 ± 4.6 | 33.8 ± 4.6 | 34.0 ± 5.0 |
| | | | |
| Mother's ethnicity (n = 135) | | | |
| White/Caucasian | 43 | 22 | 16 |
| Black | 1 | 0 | 0 |
| Hispanic | 1 | 2 | 1 |
| Asian | 7 | 16 | 3 |
| South Asian | 3 | 4 | 4 |
| Indigenous/First Nations | 0 | 2 | 0 |
| Other | 1 | 6 | 3 |
| | | | |
| Parity (n = 138) | | | |
| Nulliparous | 36 | 23 | 24 |
| 1 | 17 | 25 | 3 |
| 2 | 2 | 5 | 1 |
| 3 | 1 | 1 | 0 |
| | | | |
| Gestational age (wks) (n = 141) | 39.1 ± 0.9 | 39.9 ± 1.4 | 40.0 ± 1.2 |
| Birth wt (g) (n = 141) | 3,438 ± 512 | 3,606 ± 512 | 3500 ± 512 |
| | | | |
| Sex (n = 141) | | | |
| Female | 31 | 32 | 18 |
| Male | 26 | 34 | 9 |
| Other | 0 | 0 | 1 |
| | | | |
| Hospital or home birth (n = 140) | | | |
| Hospital | 53 | 56 | 28 |
| Home | 3 | 0 | 0 |
| | | | |
| Maternal antibiotics at delivery (n =138) | 11 | 56 | 28 |
| Maternal fever during labor (n = 138) | 5 | 0 | 3 |
| Chorioamnionitis (n = 138) | 1 | 0 | 2 |
| Meconium staining (n = 138) | 10 | 1 | 11 |
| Vaginal seeding (n = 141) | 0 | 1 | 0 |
| Twins | 1 pair | 2 pairs | None |

[a]C/S, caesarean section.

primer trimming and quality control, 2,053,681 high-quality reads were retained for variant calling, giving a final data set of 119 meconium samples and 19 sequencing controls with at least 1 read available for analysis. These data set comprised 220 amplicon sequence variants (ASVs) matching 126 unique hits, or "nearest neighbors" (NNs), in cpnDB.

The read numbers for meconium samples were low and did not differ significantly from any negative-control (Fig. 1A; $P > 0.999$ for all comparisons), reflecting the low microbial biomass. No negative controls contained >1,000 reads (Fig. 1A, dashed line), while 20 meconium samples exceeded this threshold, with a median number of 14,004 reads/sample (range, 1,241 to 16,961). This was comparable to infant stool samples sequenced in the same runs (median, 12,899 reads/sample, range 496 to 86,067).

Of the 20 high-read-count samples, 17 (85%) had a gross appearance consistent with transitional stool (lighter colored and looser) than meconium (dark, dense, and tar-like) documented upon receipt in the laboratory, compared with none in the low-read-count group. Clinical variables potentially contributing to higher read counts were interrogated by multiple regression analysis; only time from birth to sample collection was significantly associated (Table 2; $P < 0.01$, $R^2 = 0.137$). Consistent with regression analysis, time from birth to sample collection was higher in samples with >1,000 reads (median, 30.0 h versus 18.3 h; Mann-Whitney U, $P < 0.01$), though collection time was not significantly correlated with read numbers overall ($r^2 = 0.117$, Spearman's rank; $P = 0.434$) (Fig. S2A and B). No significant correlation between the

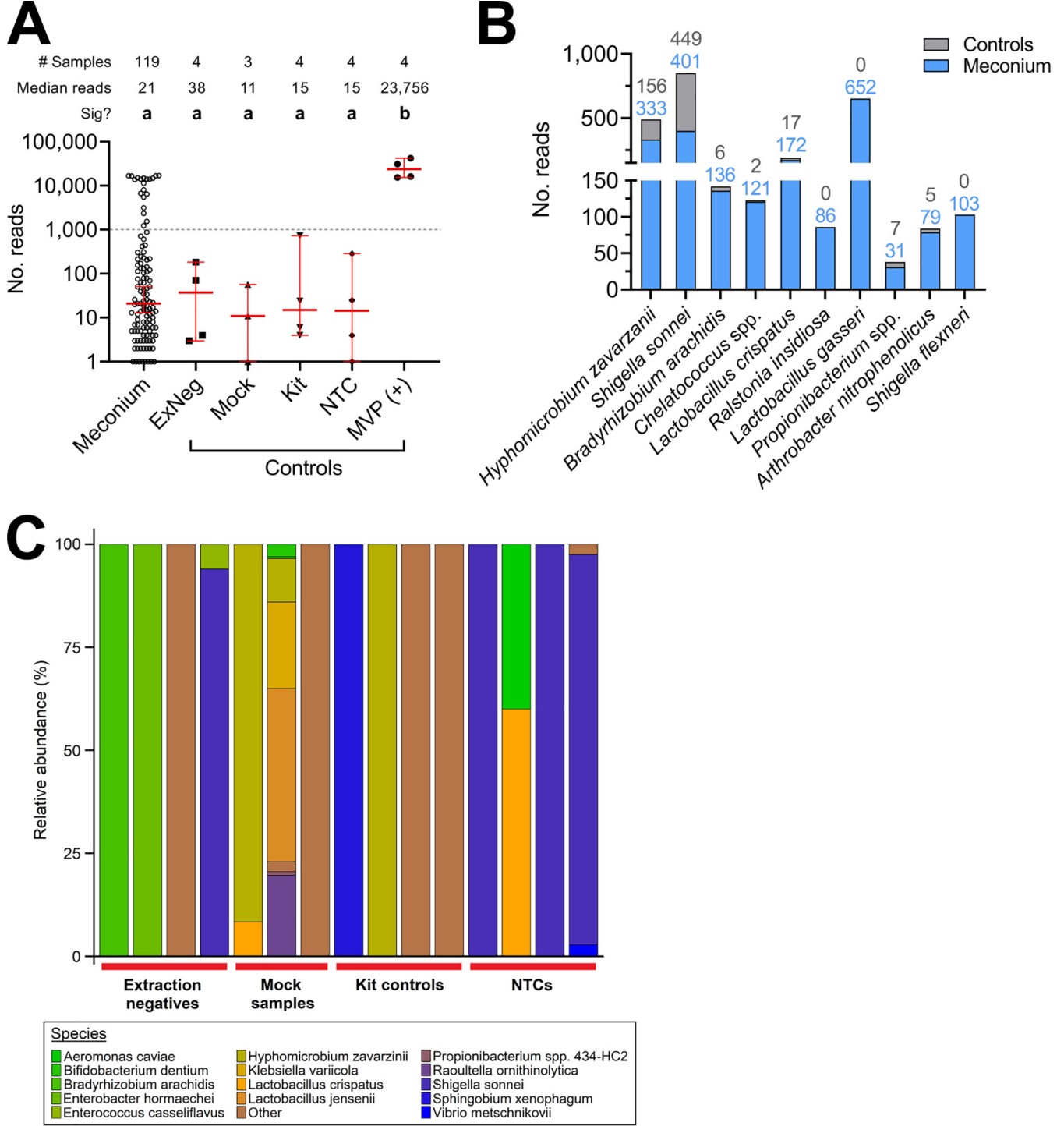

FIG 1 Meconium harbors negligible microbial content. (A) Number of *cpn*60 reads obtained from meconium samples, extraction negative controls, kit controls, mock samples, NTCs (no template control), and the mixed vaginal panel positive control (MVP). Red lines indicate the median (±95% confidence interval [CI]). Only 20 meconium samples had >1,000 reads (dashed line), and meconium read numbers did not differ significantly from negative controls (a); however, read counts from the MVP positive control were significantly higher than those for all other sample types (b; Kruskal-Wallis test with Dunn's multiple-comparison correction, $P < 0.01$, $Z = 3.23$). (B) Total read counts of the 10 most frequently detected NNs across the data set, excluding samples with >1,000 reads (colored numbers indicate total read counts in controls and meconium). Comparable read numbers in meconium and controls illustrate the issue of contamination when studying low-microbial-biomass samples. (C) Species-level profiles of sequencing negative controls showing the contamination introduced during the sequencing workflow. Any genera representing <5 reads across all control samples were collapsed into "other."

**TABLE 2** Multiple logistic regression analysis of clinical variables versus *cpn*60 read count classification (more or fewer than 1,000 reads) for 118 samples for which all variable data were available (McFadden's pseudo-$R^2$ = 0.137)[a]

| Variable | OR | 95% CI | Sig? |
|---|---|---|---|
| Delivery mode (vs elective C/S) | | | |
| Vaginal delivery | 0.52 | 0.13–2.00 | NS |
| Emergent C/S | 1.80 | 0.29–14.94 | NS |
| | | | |
| Membrane rupture length | 0.98 | 0.94–1.02 | NS |
| Birth-sample collection interval | 0.95 | 0.92–0.99 | $P < 0.01$ |
| Meconium staining | 0.59 | 0.14–2.65 | NS |

[a]C/S, caesarean section; OR, odds ratio; sig, significant; NS, not significant.

duration of ruptured membranes and read numbers was detected ($R^2 = 2 \times 10^{-5}$, Spearman's rank; $P = 0.588$), and rupture duration was not significantly different between the 20 high-read samples and those with $<$1,000 reads (median, 11.9 h versus 8.5 h; Mann-Whitney U, $P = 0.454$) (Fig. S3A and B). The distribution of delivery mode frequencies (vaginal versus elective caesarean versus emergency caesarean; Table 1) was not significantly different between samples with $>$1,000 and those with $<$1,000 reads (Chi-square test, $P = 0.153$), nor was that of meconium staining (6/20 versus 16/121; Chi-square test, $P = 0.113$).

After collapsing ASVs into unique NNs, species richness across all meconium samples was low, with a median of 1 NN per sample detected in both meconium (range 1 to 9) and negative controls (range 1 to 12). A total of 61 meconium samples (50.4%) and 9 controls (47.4%) contained reads corresponding to only a single NN, while 113 meconium samples (93.4%) and 13 controls (68.4%) were dominated by a single NN ($>$50% relative abundance) (Data set S1). The 20 samples showing higher read numbers were all dominated by a single NN, corresponding in many cases to enteric species such as *Escherichia coli* (often misclassified as *Shigella* spp. due to high *cpn*60 sequence similarity [30]), *Enterococcus faecalis*, *Bifidobacterium longum*, and *Bacteroides vulgatus* (Fig. S4). The microbial composition of this subset of meconium samples closely matched that of stool from 3-month-old infants also enrolled in the Legacy study (Dos Santos SJ, Hill JE, Money DM, Maternal Microbiome LEGACY Project Team, unpublished data). Many taxa frequently detected in meconium samples were also detected within negative controls at similar levels (Fig. 1B and C; Data set S1). For example, sequences corresponding to *Hyphomicrobium zavarzanii*, *Chelatococcus* spp. and *Bradyrhizobium arachidis* (commonly found in soil and water [31, 32]) are likely exogenous contaminants rather than truly of neonatal origin. This was exemplified by low median read numbers for the most frequently detected NNs across all meconium and sequencing controls (Table S2). In line with this, no taxa were found to be differentially abundant in meconium samples compared to sequencing negative controls using the ALDEx2 package ($P > 0.831$ for all taxa).

In terms of microbial composition, meconium samples and negative controls could not be distinguished. Principal-component analysis (PCA) showed no significant clustering by sample type (permutational multivariate analysis of variance [PERMANOVA]; Fig. 2A; $P = 0.923$, $r^2 = 0.02$); however, a statistically significant effect was observed for delivery mode (Fig. 2B; $P = <0.05$, $r^2 = 0.03$). PERMANOVA is sensitive to group differences in data dispersion and can erroneously reject the null hypothesis of no difference if there is significant heterogeneity, especially in unbalanced study designs (33). Testing for the latter revealed significant differences in dispersion between delivery modes (permutational analysis of multivariate dispersions [PERMDISP]; $P < 0.05$; see Discussion). Similarly, initial statistical testing suggested a significant effect of extraction kit batch number on sample clustering (Fig. 2C) (PERMANOVA; $P < 0.001$, $r^2 = 0.03$), though a significant difference in data dispersion was subsequently reported (PERMDISP; $P < 0.01$). This heterogeneity was driven by the absence of high-read-number meconium samples among those extracted using a kit from one of the three different batch numbers used (Fig. 2C, blue points). This demonstrates,

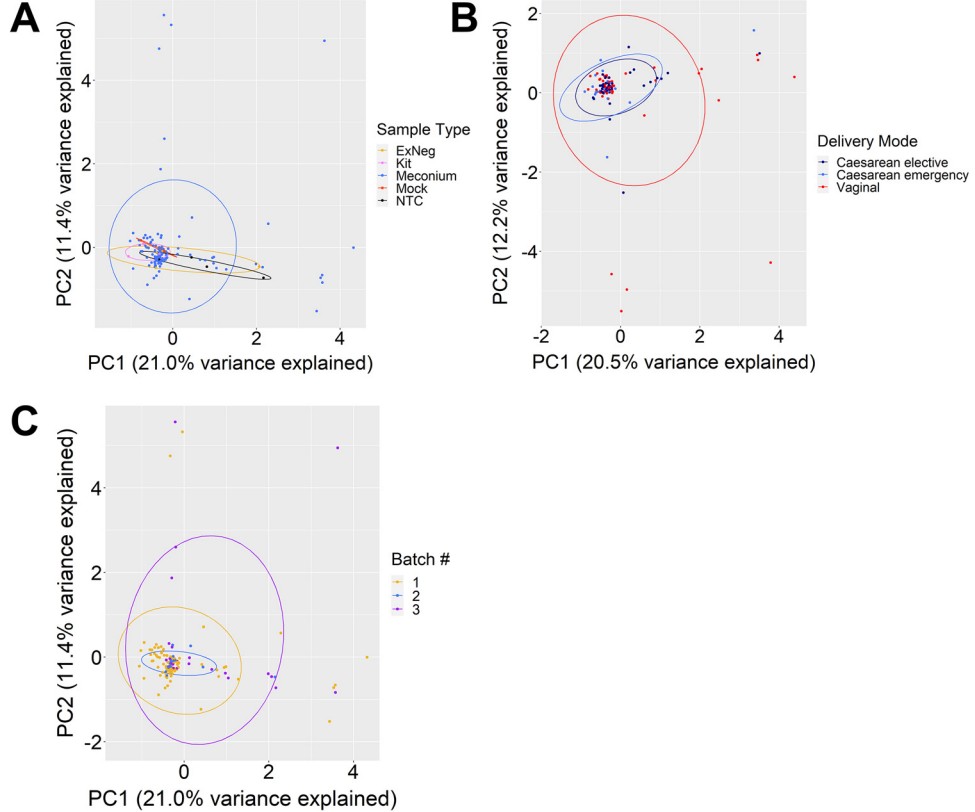

**FIG 2** Microbiome profiles of meconium show no compositional differences compared to sequencing negative controls and are not affected by delivery mode or sequencing batch. Principal-component analysis (PCA) of log ratio transformed cpn60 read count data grouped by sample type, delivery mode, and extraction kit batch number. (A) PERMANOVA reported no significant separation by sample type across the entire data set ($P = 0.923$, $r^2 = 0.02$) and no significant differences in between-group dispersion ($P = 0.246$, $F = 1.33$). (B) Meconium samples were reported to cluster significantly by delivery mode ($P < 0.05$, $r^2 = 0.03$); however, permutation of dispersion data also showed a significant difference in between-group dispersion ($P < 0.05$, $F = 4.76$). (C) Samples across the entire data set also clustered significantly by sequencing run ($P < 0.001$, $r^2 = 0.03$) with significant differences in dispersion between groups ($P < 0.01$, $F = 4.23$). Lack of evident separation of samples by delivery mode and extraction kit batch number on the respective plots, combined with permutation analysis of dispersion data, indicates that the "significant" clustering results should be interpreted as statistical artifacts rather than true biologically important differences.

therefore, that the significant results returned by PERMANOVA for delivery mode and extraction batch represent statistical artifacts rather than real biological differences in microbiome composition due to delivery mode and batch effects.

**Estimation of total bacterial load in meconium.** The inability to distinguish meconium samples and negative controls by cpn60 microbiome profiling led us to estimate total bacterial abundances in meconium via quantitative PCR (qPCR) targeting the 16S rRNA gene. Infant stool samples from 3-month-old infants were simultaneously assayed for comparison. No significant difference between mean threshold cycle ($C_T$) values of meconium (27.2 ± 2.8, $n = 139$) and negative-control samples (27.8 ± 0.5, $n = 11$) was observed (Kruskal-Wallis test with Dunn's multiple-comparison correction; $P = 0.582$); however, the mean $C_T$ value for infant stool samples (16.2 ± 2.8) was significantly lower ($P < 0.0001$) than both meconium and controls (Fig. 3). Furthermore, copy numbers of the 16S rRNA gene per gram differed by approximately 4 to 5 orders of magnitude between meconium and infant stool (Fig. 3B). Finally, we compared read numbers with 16S rRNA $C_T$ values for the 20 samples with >1,000 reads and found that these samples had significantly lower $C_T$ values than those of to the remaining 97 (Mann-Whitney U test; $P < 0.001$) and showed a significant inverse correlation with read numbers ($r^2 = 0.327$, Spearman's rank; $P < 0.05$) (Fig. S5).

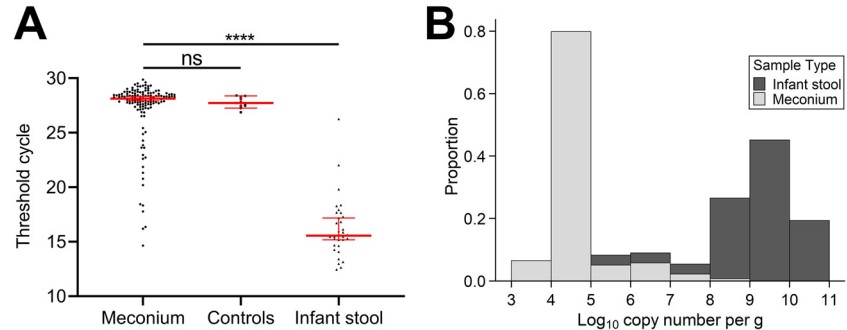

**FIG 3** Total bacterial loads of meconium samples are generally comparable to negative sequencing controls. (A) Total bacterial loads in meconium ($n = 137$) were estimated via 16S rRNA qPCR and compared to those of sequencing negative controls ($n = 11$) and stool from 3-month-old infants ($n = 31$). Data points represent the mean of duplicate reactions, while red lines indicate the median ($\pm 95\%$ CI). No significant difference was seen between cycle thresholds of meconium and negative controls (Kruskal-Wallis test with Dunn's multiple-comparison correction, $P = 0.852$, $Z = 1.07$); however, meconium exhibited a significantly lower cycle threshold than infant stool ($P < 0.0001$, $Z = 8.65$). (B) Starting quantities of the 16S rRNA gene in meconium and infant stool were expressed as $\log_{10}$ copy number per gram of material (mean from duplicate reactions) and plotted as a histogram. Copy numbers in the majority of meconium samples were approximately 4 to 6 orders of magnitude lower than those in infant stool.

**Culture of viable organisms from meconium.** We screened meconium samples for the presence of viable bacteria using culture medium that supports the growth of multiple organisms commonly detected in our microbiome data set. Aerobic culture on CHROMagar Orientation medium yielded 101 isolates from 59/141 samples (41.8%), 44 (43.6%) of which were coagulase-negative staphylococci that are recognized primarily as healthy skin microbiota (Table 3). No growth was detected on phosphate-buffered saline (PBS) control plates, while culture-positive plates inoculated with undiluted meconium (see Materials and Methods) grew only a few colonies under aerobic conditions (Fig. S6). The most commonly identified species were *Staphylococcus epidermidis* ($n = 25$), *Enterococcus faecalis* ($n = 24$), and *Escherichia coli* ($n = 12$). No significant difference in isolate distribution was observed when grouping by read count ($>1,000$ versus $<1,000$ reads; Chi-square test, $P = 0.928$, $P = 0.102$, and $P = 0.248$, respectively).

Finally, we investigated if these strains represent clonal lineages (which could indicate that isolates originated from a common source) or different strains (which could indicate that the cultured isolates are representative of each infant's microbial environment). Pulsed-field gel electrophoresis (PFGE) revealed extensive strain variation between isolates of *E. coli* and *S. epidermidis* (Fig. 4A and B). Only one pair of *E. coli* isolates could not be differentiated based on Dice similarity coefficients (Table S3), while two *S. epidermidis* isolates originating from a pair

**TABLE 3** Species identification of 84 isolates cultured from meconium, as defined by *cpn*60 Sanger sequencing and comparison to cpnDB[a]

| *cpn*60 ID | No. of isolates |
| --- | --- |
| *Citrobacter* spp. | 1 |
| *Enterococcus faecalis* | 24 |
| *Enterococcus faecium* | 1 |
| *Escherichia coli* | 12 |
| *Klebsiella pneumoniae* | 1 |
| *Staphylococcus epidermidis* | 25 |
| *Staphylococcus haemolyticus* | 2 |
| *Staphylococcus hominis* | 5 |
| *Staphylococcus intermedius* | 2 |
| *Staphylococcus lugdunensis* | 3 |
| *Staphylococcus pasteuri* | 2 |
| *Staphylococcus warneri* | 5 |
| *Streptococcus parasanguinus* | 1 |

[a]A total of 17/101 isolates repeatedly failed *cpn*60 PCR and were not identified.

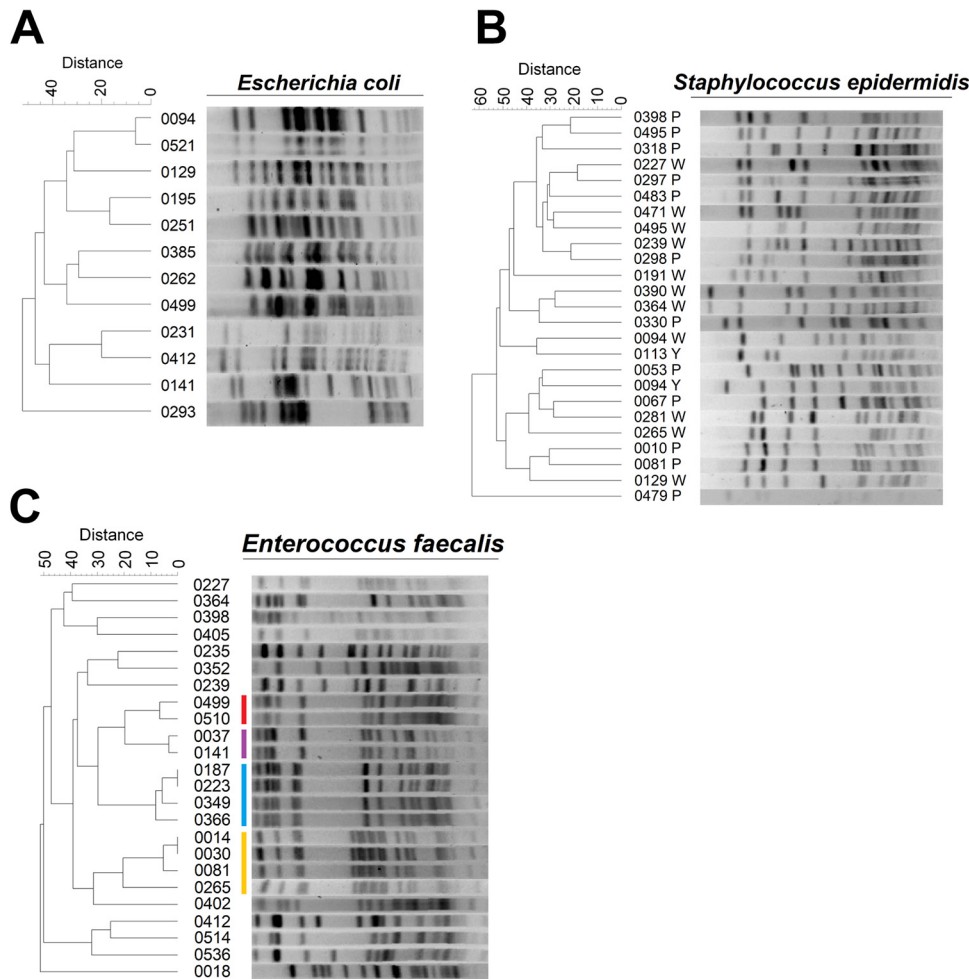

**FIG 4** Strain differences are present and abundant among the top three species cultured from meconium. DNA was extracted from overnight cultures of the three most prevalent species isolated from meconium and subjected to restriction digest prior to pulsed-field gel electrophoresis. Dendrograms of banding profiles were constructed using UPGMA based on Dice similarity coefficients. (A and B) The similarity between digest patterns of *Escherichia coli* (A) and *Staphylococcus epidermidis* (B) was generally low, indicating nonrelated strains. (C) Isolates of *Enterococcus faecalis* showed four evident clusters of indistinguishable banding patterns, suggesting the presence of multiple identical strains (colored bars).

of twins were classed as "related" (Table S4). Conversely, only 12/24 (50%) *E. faecalis* isolates were considered unique by similarity coefficients (Table S5). The remaining isolates clustered into two pairs (one related and one identical) and two indistinguishable tetrads (Fig. 4C and Table S5). Four clusters of related enterococcal isolates originated from infants delivered, in some cases, within days of each other, raising the possibility of hospital-associated acquisition (Table S6).

## DISCUSSION

Using multiple approaches, we highlighted the difficulty in identifying a distinct meconium microbiome against a background of widespread, exogenous DNA contamination. Thus, our sequencing data do not support the existence of a bona fide microbiome within meconium and, while we were able to isolate low levels of bacteria from some samples, these more likely represent contamination from skin or diapers than *in utero* colonization (34). We opted for the *cpn*60 universal barcode over the commonly used 16S rRNA gene, as it provides superior resolution in discriminating between bacterial taxa (35, 36), the latter typically limiting identification to the genus level. Subspecies and strains of multiple species can be differentiated by their *cpn*60 sequence, while 16S rRNA

sequencing fails to unambiguously distinguish between them. Indeed, analyses have proven it to be a more suitable molecular barcode according to the framework for evaluating candidate genes set out by the International Barcode of Life project, with far greater sequence heterogeneity between closely related taxa (35). It is also a particularly valuable approach for assessing the vaginal microbiome and corresponding infant microbiome due the limited diversity of genera and need for species- and subspecies-level discrimination.

Several groups have illustrated the presence and stochastic nature of bacterial DNA in nucleic acid extraction kits (20, 22). Accordingly, best practices for microbiome studies now recommend inclusion of multiple controls throughout the sequencing workflow (21). However, many studies reporting detection of unique microbial communities in the placenta, amniotic fluid, or meconium failed to include such controls routinely or only assessed their DNA content by gel electrophoresis, despite the fact that DNA can be detected by sequencing even when no bands are present (11–15, 17–19, 23). In this context, no confident distinction can be made between contaminants and taxa truly present in such environments. When controls are included, contamination becomes evident; one meconium study detected thousands of reads aligning to *Pelomonas puraquae*, a species isolated from industrial water and associated with environmental pollutants in marine studies (37, 38), in 38/43 samples and within negative controls. Previous work has identified *Pelomonas* spp. as contaminants of extraction kits and ultrapure water (20, 22).

Recent large-scale investigations report a high degree of overlap between placental operational taxonomic units (OTUs) and known contaminants (25–28). In our study, the similarity of read numbers and total bacterial loads, as well as compositional overlap between meconium samples and negative controls, further underscored the abundance of contamination within this data set (Fig. 1 and 3). Contaminating *cpn*60 sequences were not uniformly distributed across all controls, suggesting sporadic, rather than ubiquitous, contamination across the sample set. This pattern complicates data interpretation such that one cannot exclude the possibility that the several hundred reads of taxon X detected in sample Y were not a result of contamination, despite its lack of detection within any negative control. This is underscored by the absence of any differentially abundant taxa identified by ALDEx2 (Data set S1). Other studies of the microbial content of meconium have, despite their aforementioned limitations, detected similar taxa to our own study; *Enterococcus*, *Escherichia*, *Propionibacterium*, *Bacteroides*, and other genera of the *Enterobacteriaceae* are among the most commonly reported taxa identified in meconium (12, 15, 16, 39).

Delivery mode exerts a strong effect on infant gut microbiome composition (7, 8), though conflicting data exist for meconium (13, 15, 23); however, it remains possible that subtle differences in founding populations influence microbial succession and community structure in early life (40). In our study, samples separated significantly by delivery mode; however, this is likely to be statistically, but not biologically, significant. Many taxa detected in meconium likely represent exogenous contaminants, and there is considerable overlap between groups on PCA plots (Fig. 1 and 2). Likewise, significant differences in data dispersion between groups can cause PERMANOVA to erroneously reject a null hypothesis of no difference (33). Therefore, while we can be confident about the lack of difference by sample type ($P = 0.923$, null hypothesis accepted), we caution against interpretation of our data as affirming previous reports that delivery mode impacts meconium microbiome composition. Similarly, some studies describe differing microbiome compositions based on exposure to antimicrobial agents (41). In our study, all mothers undergoing caesarean delivery received antimicrobial prophylaxis prior to surgery; thus, delivery mode represented a *de facto* proxy for antimicrobial exposure during labor and delivery. Despite this, permutation analysis of our microbiome data by delivery mode (Fig. 2C), as well as incredibly low estimates of total bacterial abundances for most meconium samples regardless of delivery mode (Fig. 3), suggest that antibiotic exposure did not affect our detection of a meconium microbiome.

The largest determinant affecting detection of microbial DNA was sample appearance. Of 20 samples with higher read numbers, 17 resembled transitional stool but were included in the study due to their collection within 72 h of birth. Previous studies of neonatal gut

transit time estimate up to 3 days before meconium is completely expelled but note the gradual transition from dark, viscous meconium to looser, yellow stool as feeding begins (42). It is entirely possible that these samples contain stool rather than meconium, providing the most likely explanation for their higher read numbers. Alternatively, there was a longer interval between birth and meconium collection for these samples compared to those with <1,000 reads, and 12/20 of these samples were collected after the first 24 h of life, after which time, microbial colonization of the gut has already likely begun (43). Colonization between membrane rupture and birth, however, appeared minimal in our study given the absence of correlation between this interval and total bacterial counts, as documented by others (16).

Like others studying neonatal samples (12, 39), we cultured multiple species from meconium, particularly, coagulase-negative staphylococci (Table 3). However, the latter are common skin commensals that are well adapted to growth *in vitro* and could represent contamination from perineal skin during sample collection, similar to sterile blood culture (34). This was reinforced by PFGE data showing distinct strains rather than clones of a few systematic contaminants (Fig. 4B). The clonal clusters of *E. faecalis* strains we observed may be attributed to acquisition from a common hospital environment, given the close distribution of delivery dates within each cluster (Table S6). Multiple strains of *E. faecalis* circulate in and among hospitals worldwide, with many strains being stably maintained within a single hospital over many years (44). Finally, we also isolated distinct strains of *E. coli* (Fig. 4A), though it is not possible from our data to confirm or rule out residence of these bacteria within the fetal gut; hence, postpartum colonization remains a plausible explanation. While the use of a single set of culture conditions can be considered a limitation in our study, the purpose of attempting cultivation was not to compile an exhaustive list of species present in meconium but, rather, to determine the presence of any viable organisms. Likewise, the long-term storage (~1 year) of samples at –80°C prior to culture may also have reduced bacterial viability. Despite this, our culture data still imply that the origin of most isolates recovered from meconium lies outside the fetal environment.

Overall, we have demonstrated that meconium does not harbor a resident microbial community that can be distinguished from the exogenous contamination introduced during sample processing and the sequencing workflow. Our data further illustrate the requirement for the incorporation of robust controls in future sequencing studies investigating environments with inherently low microbial biomass. In line with recent studies addressing contamination in placental and amniotic fluid microbiomes, this study provides strong evidence that the gut microbiome is not established prior to birth.

## MATERIALS AND METHODS

**Study population.** This study subset forms part of a prospective, longitudinal cohort study of pregnant women delivering at term and the role of the mother's vaginal microbiome in predicting the infant gut microbiome composition (Maternal Microbiome Legacy Project). The present study included pregnant women >18 years of age (expecting to deliver in hospital or at home) recruited from the British Columbia Women's Hospital + Health Centre (BCWH) in Vancouver, Canada. Informed consent for study participation was obtained and ethical approval was granted by the University of British Columbia (UBC) Children's and Women's Research Ethics Board (H17-02253). Participants were excluded based on the following criteria: inability to give informed consent, participation in drug or probiotic trials, gestational age <37 weeks at delivery, known major fetal anomalies, triplet or greater gestation, placenta previa at delivery, placental abruption, and emergency intrapartum complications. All clinical metadata were collected via interview or review of medical charts and managed using Research Electronic Data Capture (REDCap) tools hosted at BC Children's Hospital Research Institute (45).

**Sample collection.** Meconium, defined as the earliest intestinal discharge passed within 72 h of birth, was collected from each infant. Infant stool samples were similarly obtained at 3 months of age (±2 weeks); samples were self-collected by parents and kept in sealed bags in a refrigerator until collection by the study team and were transported to the lab in a cooler. Meconium and stool were scraped from diapers using sterile spatulas in a biosafety cabinet, aliquoted into cryovials, and stored at –80°C. "Mock sample" controls were generated by handling three empty cryovials in the same manner to capture any contaminants introduced during the process. Samples were shipped on dry ice to the University of Saskatchewan and immediately stored at –80°C.

**DNA extraction and negative controls.** Total genomic DNA was extracted from 200 mg of meconium or infant stool using a commercial kit (MagMax DNA Ultra v2.0; Applied Biosystems) on a KingFisher Flex platform (Thermo Fisher Scientific, Waltham, MA, USA). Extractions were performed on different days to prevent cross-contamination, and batch numbers of the extraction kits were noted. Several negative controls were also included in line with recent recommendations for microbiome studies (21). Extraction-negatives were included with every DNA extraction (400 $\mu$l molecular biology-grade water in place of sample), to evaluate contamination introduced from the laboratory environment during the extraction. DNA was also extracted from the mock sample controls described above in order to identify contamination arising during sample processing. Kit reagents underwent DNA extraction separately from samples to assess contaminating DNA in the extraction kits.

**PCR.** Amplification and indexing of the *cpn*60 barcode region were performed as previously described (46) (Table S1). A mixture of 20 plasmids containing *cpn*60 sequences from constituents of the vaginal microbiome (47) (mixed vaginal panel, MVP) mixed in equal proportions was employed as a positive control to confirm successful amplification and sequencing. The presence of amplifiable DNA in meconium samples was confirmed by amplifying a 341-bp region of the human mitochondrial cytochrome *c* oxidase I gene (Table S1).

**Sequencing and analysis.** Indexed amplicons were quantified, normalized, and pooled as previously described (46). Diluted 10-pM libraries containing 5% PhiX DNA were sequenced on an Illumina MiSeq platform using a 500-cycle reagent kit v2 (401 R1, 101 R2). Only R1 sequences were used for analysis. Sequencing primers were removed using Cutadapt (48), and quality trimming was performed with Trimmomatic (49) (minimum length, 150 bp; minimum average Phred score, Q30). Variant calling was carried out in the QIIME2 package using DADA2, truncating to 250 bp from the 5′ end (30, 50, 51). Amplicon sequence variants (ASVs) were aligned to a nonredundant version of cpnDB using wateredBLAST (52, 53). Only ASVs with >55% sequence similarity to a hit in cpnDB were retained for further analysis (54). The resulting feature table was collapsed into nearest neighbors (NNs), whereby ASV frequencies aligning to the same cpnDB entry were combined. All sequence data were deposited in the NCBI Sequence Read Archive (BioProject number PRJNA665246).

**16S rRNA qPCR.** Total bacterial loads were estimated using a qPCR assay targeting the 16S rRNA gene as described previously (55) (Table S1). Reactions were assayed in duplicate alongside positive and no-template controls, and the copy number per gram of sample was calculated.

**Culture and isolate identification.** A loopful of meconium was suspended in 1 ml PBS, and 100 $\mu$l of resuspended sample was plated on CHROMagar Orientation agar (CHROMagar, Paris, France) before incubation at 37°C for 24 h. Samples with >1,000 reads were serially diluted in PBS down to $10^{-4}$ to avoid confluent growth. PBS used for suspension of meconium was also plated to check for contaminants. All unique colonies were subcultured for creation of glycerol-milk freezer stocks (20% vol/vol glycerol, 4% wt/vol skim milk powder, 1% wt/vol glucose) and identification by *cpn*60 barcode PCR and sequencing (56) (Table S1).

**Pulsed-field gel electrophoresis (PFGE).** PFGE of *Escherichia coli*, *Enterococcus faecalis*, and *Staphylococcus epidermidis* isolates was performed using a CHEF-DR III system (Bio-Rad Laboratories, Inc., Hercules, CA, USA) according to standardized CDC protocols (57). Premade 10× Tris-borate-EDTA (TBE; Sigma-Aldrich, St. Louis, MO, USA) was diluted to 0.5× before use in electrophoresis. Agarose gels were imaged on a GelDoc XR+ system (Bio-Rad Laboratories Inc.) set to 6-s exposure under UV illumination. Images were analyzed with GelComparII (bioMérieux, Marcy-l'Étoile, France), using unweighted pair group method using average linkages (UPGMA) of Dice similarity coefficients for clonality assessment.

**Statistical analysis.** Analyses were performed in Prism v9.01 (GraphPad Software, San Diego, CA, USA) or RStudio v4.0.3. Feature tables containing read counts for stool and meconium microbiome profiles were center log-ratio (CLR) transformed using ALDEx2 (58) (aldex.clr). Only NNs present in at least 2% of samples were retained for analysis. CLR-transformed data were analyzed by principal-component analysis (PCA; prcomp) to visualize compositional similarity. The vegan package (59) was employed to test for differences in microbiome composition (adonis/PERMANOVA) and data dispersion (33) (betadisper/permutest). The aldex wrapper function was used to assess differential abundance between microbiome profiles of meconium and negative controls (58).

## SUPPLEMENTAL MATERIAL

Supplemental material is available online only.

**SUPPLEMENTAL FILE 1**, XLSX file, 0.2 MB.
**SUPPLEMENTAL FILE 2**, PDF file, 0.6 MB.

## ACKNOWLEDGMENTS

We gratefully acknowledge Iveoma Udevi, Beheroze Sattha, Melissa Watt, Lilija Berngards, Nimrat Binning, and Sukhpreet Buttar for their hard work and assistance with participant recruitment, sample and data collection, and data entry and the nursing staff at BC Women's Hospital for their assistance with recruitment and sample collection. We thank Champika Fernando for her help with lab work and microbiome profiling, and Michelle Sniatynski for her guidance with PFGE experiments. We are most grateful to all of the participants of the Maternal Microbiome Legacy Project for taking the time and

effort to participate in this study. Finally, we also thank the reviewers for their constructive feedback on the manuscript.

This work was funded by a Canadian Institute of Health Research grant and was virtually presented at the annual meeting of the Infectious Diseases Society for Obstetrics and Gynecology in August 2020.

The Maternal Microbiome Legacy Project Team is Deborah M. Money, Janet E. Hill, K.S. Joseph, Julie E. van Schalkwyk, Arianne Y.K. Albert, Chelsea N. Elwood, Soren Gantt, Kirsten Grabowska, Jennifer A. Hutcheon, Matthew G. Links, Amee R. Manges, Sheona M. Mitchell-Foster, Tim J. Dumonceaux, Zoë G. Hodgson, Janet Lyons.

We declare no conflicts of interest.

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
