## [Reviewer comments · Microbiology Spectrum]

Microbiology Spectrum

Early neonatal meconium does not have a demonstrable microbiota determined through use of robust negative controls with *cpn60*-based microbiome profiling.

Scott Dos Santos, Zahra Pakzad, Chelsea Elwood, Arianne Albert, Soren Gantt, Ameer Manges, Tim Dumonceaux, Evelyn Maan, Janet Hill, Deborah Money, and The Maternal Microbiome Legacy Project Team

Corresponding Author(s): Deborah Money, University of British Columbia

Review Timeline:

Submission Date:	May 3, 2021
Editorial Decision:	July 23, 2021
Revision Received:	August 19, 2021
Accepted:	August 26, 2021

Editor: Zhenjiang Xu

Reviewer(s): Disclosure of reviewer identity is with reference to reviewer comments included in decision letter(s). The following individuals involved in review of your submission have agreed to reveal their identity: Tao Ding (Reviewer #1)

Transaction Report:

DOI: <https://doi.org/10.1128/Spectrum.00067-21>

July 22, 2021

Dr. Deborah M. Money
University of British Columbia
Department of Obstetrics and Gynaecology
Vancouver, BC
Canada

Re: Spectrum00067-21 (Early neonatal meconium does not have a demonstrable microbiota determined through use of robust negative controls with *cpn60*-based microbiome profiling.)

Dear Dr. Deborah M. Money:

I have received the reviews of your manuscript. While your study addresses an interesting question, the reviewers raised several good questions on the text and analyses. Therefore, I invite you to respond to the reviewers' comments and revise your manuscript. In particular, please note the following.

Thank you for submitting your manuscript to Microbiology Spectrum. When submitting the revised version of your paper, please provide (1) point-by-point responses to the issues raised by the reviewers as file type "Response to Reviewers," not in your cover letter, and (2) a PDF file that indicates the changes from the original submission (by highlighting or underlining the changes) as file type "Marked Up Manuscript - For Review Only". Please use this link to submit your revised manuscript - we strongly recommend that you submit your paper within the next 60 days or reach out to me. Detailed information on submitting your revised paper are below.

Link Not Available

Sincerely,

Zhenjiang Xu

Journals Department
Reviewer comments:

Reviewer #1 (Comments for the Author):

The authors need to consider the following comments to improve the quality of this manuscript before it is suitable for publication.

1. Why did the authors choose cpn60-based amplicon sequencing to define the microbial composition of meconium instead of the 16S rRNA gene sequencing? Both are available options, but the author should explain their rationale.
2. Line 167, "Only ASVs with >55% sequence similarity to a hit in cpnDB were retained for further analysis". This similarity threshold 55% is too low. What's the reason to use this number?
3. Figure 1B shows that many species such as *Lactobacillus crispatus* have 0 reads in the control group and a higher reads numbers in the meconium sample. This result seems to be the exact opposite of what the authors intended to prove. The authors need to explain this before the conclusion of this manuscript is believed to be valid.
4. Line 261-275, in the content of this paragraph, the author seems to have realized that the PCA method is not ideal for the display of differences in the sparse composition table. Is it possible to adopt a more appropriate method to display the results of this part?
5. Figure 4, Similarity between digest patterns of *Escherichia coli* and *Staphylococcus epidermidis* was generally low, indicating non-related strains. Isolates of *Enterococcus faecalis* showed four evident clusters of indistinguishable banding patterns, suggesting the presence of multiple identical strain. These results seem to indicate that the bacteria in meconium may be partly from environmental pollution and partly from maternal origin. I don't think such findings support the authors' conclusions.

Reviewer #2 (Comments for the Author):

In this article, Scott J. Dos Santos et al. aimed at exploring the microbial communities in early neonatal human meconium, using cpn60 based microbiome profiling of 141 neonatal meconium samples. They also compared the bacterial loads by performing qPCR for 16S rRNA on neonatal meconium, infant stool and controls. Based on their observations the authors conclude that neonatal meconium is devoid of any microbial presence. There are certain aspects mentioned below which need further investigation and clarification:

1. Although the authors have taken "Mock sample" controls generated by handling empty cryovials in similar manner, there are no controls taken in the study before the transfer of samples to cryovial. It is important to have control swabs for each case to have an accurate comparison between case and control. Currently there are only 4 data-points in each control group, probably reflecting the batch controls. Also, there seems to be a clear difference between read-number of 50% of ExNeg samples which may lead to misleading interpretations in dearth of sufficient number of controls.
2. The authors must provide quantifications of β -diversity to compute community dissimilarity scores instead of directly plotting the read-numbers. Computing Bray-Curtis Dissimilarity, Weighted UniFrac distance or any other similar approach would be an accurate demonstration of dissimilarity

among case and controls.

Authors should also give an assessment of alpha-diversity by comparing the taxonomic dissimilarity among the meconium samples themselves, to address the possibility of common contaminants originating from reagents, Hospital contaminations etc as mentioned by the authors.

3. The authors should discuss the bacterial genera identified in their analysis especially for the samples above 1000 read-counts and compare them with the genera identified in infant stool samples. If the apparent increase in microbial reads is indeed due to the time spent post birth, it would be interesting to see if they match the profile of an infant stool.

4. Did the authors normalize the weight of meconium and stool for each sample for qPCR? What was the exact weight taken for the qPCR of 16S rRNA based microbial profiling?

5. In the culture experiments, it is not clear if the samples were inoculated freshly or post cryo-storage? If the samples are not freshly inoculated, most of the bacterial taxa won't survive post cyro-storage and what authors may observe is a selective enrichment of certain bacterial species that may survive cryo-preservation or have been introduced during culturing. Additionally, were there appropriate PBS controls taken during inoculation? If yes their culture profile should also be discussed.

6. Authors have only provided aerobic culture environment to the isolates, which is suitable for the infant stool samples but not for culturing the microbes from meconium samples which would be of fetal origin. The oxygen supply to the fetal compartment during gestation is quite limited and changes through gestational timeline. Selective enrichment in aerobic cultures will result in the loss of anerobic microbes likely present in fetal meconium.

7. How do authors explain 'No significant difference in isolate distribution was observed when grouping by read count' in their culture conditions? If the samples with >1000 READS are indeed due to meconium samples being of later time points (as explained by the meconium appearance) or are contaminated, why they did not yield significantly high viable microbes in culture?

8. Authors must also discuss in their study some of the previous reports where human meconium has been associated with detectable microbial presence. Some of them are mentioned below. Also they should discuss some new reports on the presence of microbes in human fetal gut.

Stinson, Lisa F., et al. "The not-so-sterile womb: evidence that the human fetus is exposed to bacteria prior to birth." *Frontiers in microbiology* 10 (2019): 1124.

He, Qiuwen, et al. "The meconium microbiota shares more features with the amniotic fluid microbiota than the maternal fecal and vaginal microbiota." *Gut Microbes* 12.1 (2020): 1794266

Ardissone, Alexandria N., et al. "Meconium microbiome analysis identifies bacteria correlated with premature birth." *PloS one* 9.3 (2014): e90784.

Gosalbes, M. J., et al. "Meconium microbiota types dominated by lactic acid or enteric bacteria are differentially associated with maternal eczema and respiratory problems in infants." *Clinical & Experimental Allergy* 43.2 (2013): 198-211

Hu, Jianzhong, et al. "Diversified microbiota of meconium is affected by maternal diabetes status."

Staff Comments:

Preparing Revision Guidelines

For complete guidelines on revision requirements, please see the Instructions to Authors at [link to page]. **Submissions of a paper that does not conform to Microbiology Spectrum guidelines will delay acceptance of your manuscript.**

Please return the manuscript within 60 days; if you cannot complete the modification within this time period, please contact me. If you do not wish to modify the manuscript and prefer to submit it to another journal, please notify me of your decision immediately so that the manuscript may be formally withdrawn from consideration by Microbiology Spectrum.

If you would like to submit an image for consideration as the Featured Image for an issue, please contact Spectrum staff.

Responses to reviewer comments

Reviewer #1 (Comments for the Author):

The authors need to consider the following comments to improve the quality of this manuscript before it is suitable for publication.

1. Why did the authors choose *cpn60*-based amplicon sequencing to define the microbial composition of meconium instead of the 16S rRNA gene sequencing? Both are available options, but the author should explain their rationale.

Response: Our lab has been using *cpn60* for microbiome studies for many years and has previously published a comparison of *cpn60* vs. the 16S rRNA gene in terms of its resolution and ability to confidently discriminate between closely related taxa (Links *et al.* 2012. *PLoS One*; **11**: e49755). While the 16S rRNA gene is only capable of genus-level identification at best, the *cpn60* universal barcode has superior resolution and permits species (and in some cases even sub-species) level identification. It is particularly helpful in the analysis of the vaginal microbiome due to the relative low diversity of genera but need for discrimination at the species and sub-species level to fully understand this microbial microenvironment. We assumed that the early infant's microbiome would be heavily influenced by the vaginal microbiome and hence our selection of this approach. In accordance with the above comment, we have explained the rationale behind the decision to use *cpn60* in our microbiome investigation and added the following text to the first paragraph of the discussion (lines 328-335; new manuscript) and have cited two additional studies to support the statements below:

"We opted for the *cpn60* universal barcode over the commonly used 16S rRNA gene as it provides superior resolution in discriminating between bacterial taxa, the latter typically limiting identification to the genus level. Subspecies and strains of multiple species can be differentiated by their *cpn60* sequence, while 16S rRNA sequencing fails to unambiguously distinguish between them. Indeed, analyses have proven it to be a more suitable molecular barcode according to the framework for evaluating candidate genes set out by the International Barcode of Life project, with far greater sequence heterogeneity between closely related taxa. It is also a particularly valuable approach for assessing the vaginal microbiome and corresponding infant microbiome due the limited diversity of genera and need for species and subspecies level discrimination"

2. Line 167, "Only ASVs with >55% sequence similarity to a hit in cpnDB were retained for further analysis". This similarity threshold 55% is too low. What's the reason to use this number?

Response: We thank the reviewer for this comment. We have been employing this strategy for microbiome analyses after we found that reads from *cpn60* amplicon sequencing studies with less than 55% sequence similarity to any sequence in cpnDB either map to the host genome or are bacterial in origin (i.e., non-*cpn60* bacterial reads) (see Johnson *et al.* 2015. *BMC Research Notes*; **8** (1): 253). Reads above 55% similarity to cpnDB entries represent *cpn60* from species with no close match in the database. However, in this study, we observed very few amplicon sequence variants (ASVs) of *cpn60* with a sequence similarity near this threshold. In fact, only 6/220 ASVs had sequence similarities to their closest match in cpnDB below 75%, and these accounted for just 24 reads in the entire dataset.

3. Figure 1B shows that many species such as *Lactobacillus crispatus* have 0 reads in the control group and a higher reads numbers in the meconium sample. This result seems to be

the exact opposite of what the authors intended to prove. The authors need to explain this before the conclusion of this manuscript is believed to be valid.

Response: The reviewer raises an important point regarding the nature of contamination in sequencing studies, and we thank them for this comment. In Figure 1B, there are three species with 0 reads in the controls: *Lactobacillus gasseri*, *Ralstonia insidiosa* and *Shigella flexneri*. At first glance, it certainly appears that this finding is contrary to our conclusions; however, one cannot assume that the taxa found in one's negative controls represent the absolute extent of contamination within the dataset. As mentioned in the discussion of the manuscript (lines 334-337; original manuscript), the patterns of contamination we observed appeared to be stochastic, with uneven distribution of the contaminating sequences among negative controls. This phenomenon, combined with the incredibly low read numbers seen in the overwhelming majority of the meconium samples, means that we cannot interpret the detection of a few hundred reads of (to use the example above) *Lactobacillus gasseri* to as being truly present in a sample, especially when there are negative controls (e.g. MLP-0561-K, a kit control) with comparable read numbers. Furthermore, although this manuscript focuses on a small subset of more than 2,200 samples sequenced as part of the LEGACY project. Since completing the work on the meconium microbiome, we have found *cpn60* reads aligning to *Lactobacillus gasseri* in the negative controls (DNA extraction blanks and PCR controls) of further sequencing runs of LEGACY samples. This underscores the stochastic nature of the contamination.

To provide supporting evidence that there are no taxa that differentiate meconium samples and sequencing negative controls, we have performed differential abundance analysis using the ALDEx2 package for R and demonstrated that there are no taxa that are statistically more abundant in the meconium samples than in the controls. This been added to the results (lines 265-267; new manuscript) and discussion (lines 356-357; new manuscript) sections, with an appropriate description in the methods (lines 202-204; new manuscript). We have also updated Supplementary File S1 with a tab showing the results exactly as output by ALDEx2, and a description of the code to reproduce the result in RStudio.

Methods: "The aldex wrapper function was used to assess differential abundance between microbiome profiles of meconium and negative controls."

Results: "In line with this, no taxa were found to be differentially abundant in meconium samples compared to sequencing negative controls using the ALDEx2 package ($P > 0.831$ for all taxa)."

Discussion: "This is underscored by the absence of any differentially abundant taxa identified by ALDEx2 (Supplementary File S1)."

4. Line 261-275, in the content of this paragraph, the author seems to have realized that the PCA method is not ideal for the display of differences in the sparse composition table. Is it possible to adopt a more appropriate method to display the results of this part?

Response: We thank the reviewer for this comment; however, we respectfully disagree with their interpretation of the aforementioned paragraph of the results section. PCA is a standard dimensionality reduction technique commonly used to show the maximum amount of variation attributable to given variables in complex datasets, such as those generated by microbiome studies. The issue raised by the reviewer appears to be with interpretation of the statistical testing (PERMANOVA) rather than the choice of ordination method (PCA).

We reported the results of PERMANOVA on the microbiome dataset, stratified by sample type, where there was no significant difference in microbiome composition (Fig 2A; $P =$

0.923, $r^2 = 0.02$). We then reported “significant” differences when the grouping variable was delivery mode (Figure 2B; $P < 0.05$, $r^2 = 0.03$) or extraction batch (Figure 2C; $P < 0.001$, $r^2 = 0.03$), respectively. We then described one of the limitations of PERMANOVA: the fact that it is sensitive to differences in the dispersion of the data between groups. If such dispersion differences are statistically significant (which is the case for our data, given the results of PERMDISP), the null hypothesis of no difference can be rejected in error. However, this does not make the choice of analysis unsuitable. What this means is the result must be interpreted with caution without absolute reliance on the P value as to what is a ‘real’ or biologically significant difference. This aside, PERMANOVA is considered to be one of the most robust statistical tests in microbial ecology (see Anderson, 2017, WileyStatsRef).

With regard to our data, the difference in data dispersion between groups (either between delivery modes or extraction batches) may be responsible for the “significant” result from PERMANOVA, meaning that it is an artefact of the testing rather than a true biological difference. As no significant difference in composition was observed by sample type (i.e., meconium vs. sequencing controls), the issue of group dispersion impacting the effect (or lack thereof) of sample type on microbiome composition does not apply. Dispersion differences may cause the null hypothesis to be **rejected** in error, rather than accepted.. Furthermore, given that the data represent contaminants rather than a real microbiome, any differences found would be biologically meaningless. We feel that the explicit explanation given in both the results (lines 261-275; original manuscript) and discussion (lines 342-357; original manuscript) cover this succinctly.

However, the reviewer raises an important point about the lack of clarity in this section; therefore, we have changed the wording of the results (lines 280-282; new manuscript) and discussion (lines 368-373; new manuscript) sections accordingly:

Results: “This demonstrates therefore that the significant results returned by PERMANOVA for delivery mode and extraction batch represent statistical artefacts rather than real biological differences in microbiome composition due to delivery mode and batch effects.”

Discussion: “Likewise, significant differences in data dispersion between groups can cause PERMANOVA to erroneously reject a null hypothesis of no difference(40). Therefore, while we can be confident about the lack of difference by sample type ($P = 0.923$, null hypothesis accepted), we caution against interpretation of our data as affirming previous reports that delivery mode impacts meconium microbiome composition.”.

5. Figure 4, Similarity between digest patterns of *Escherichia coli* and *Staphylococcus epidermidis* was generally low, indicating non-related strains. Isolates of *Enterococcus faecalis* showed four evident clusters of indistinguishable banding patterns, suggesting the presence of multiple identical strain. These results seem to indicate that the bacteria in meconium may be partly from environmental pollution and partly from maternal origin. I don't think such findings support the authors' conclusions.

Response: The reviewer raises an important point regarding the potential origins of the isolates cultured from neonatal meconium. While the reviewer asserts that “these results seem to indicate that the bacteria in meconium may be partly from environmental pollution and partly from maternal origin”, we respectfully disagree with the latter part of the statement. We did not present any evidence suggesting that the isolates originated from a maternal source, nor did we discuss any data regarding samples collected from mothers enrolled in the Maternal Microbiome Legacy Project.

However, it may be the case that our data indicates that some isolates may originate from a foetal/neonatal source, rather than from the mother. In this case, the reviewer is quite correct. We discussed potential sources of these isolates in the discussion section (lines 370-384; original manuscript). The abundance of coagulase-negative staphylococci cultured from meconium are most likely the result of sample collection directly from diapers. Being extremely common skin organisms, this is not at all surprising. Indeed, the PFGE data support this, with each infant harbouring different strains likely acquired from their immediate surroundings. The identical strains cultured from meconium from the single pair of twins further supports this.

We also provided a potential explanation for the presence of multiple identical strains of *Enterococcus faecalis* among our culture collection (hospital acquisition, supported by birth dates of infants within these clusters). However, as the reviewer points out and is explicitly stated in the discussion section (lines 379-381; original manuscript), we cannot confirm if the *Escherichia coli* isolates were actually resident in the foetal gut or if they were acquired postpartum. This being the case, one has to consider the culture data alongside the lack of compositional differences between meconium and sequencing negative controls and extremely low bacterial loads in meconium. These findings, plus the fact that most of the isolates grown from meconium represented normal skin flora and organisms plausibly acquired from the hospital environment, led us to conclude that “the origin of most meconium isolates lies outside of the foetal environment” (lines 383-384; original manuscript).

Reviewer #2 (Comments for the Author):

In this article, Scott J. Dos Santos et al. aimed at exploring the microbial communities in early neonatal human meconium, using cpn60 based microbiome profiling of 141 neonatal meconium samples. They also compared the bacterial loads by performing qPCR for 16S rRNA on neonatal meconium, infant stool and controls. Based on their observations the authors conclude that neonatal meconium is devoid of any microbial presence. There are certain aspects mentioned below which need further investigation and clarification:

1. Although the authors have taken "Mock sample" controls generated by handling empty cryovials in similar manner, there are no controls taken in the study before the transfer of samples to cryovial. It is important to have control swabs for each case to have an accurate comparison between case and control. Currently there are only 4 data-points in each control group, probably reflecting the batch controls. Also, there seems to be a clear difference between read-number of 50% of ExNeg samples which may lead to misleading interpretations in dearth of sufficient number of controls.

Response: The reviewer is quite correct that no controls were taken prior to the transfer of the sample from the diaper to the cryovial. As the samples were collected directly from the diaper, the "mock sample" is the earliest control we could implement to assess the impact of contamination on the composition of the meconium microbiome. The reviewer also raises an important point regarding paired sampling controls; however, it would not have been financially or logistically feasible to collect and perform DNA extraction, cpn60 and index PCRs and sequence a paired sample collection control for each of the 630 infants enrolled in the Maternal Microbiome Legacy Project.

In terms of the accuracy in comparing the composition of meconium samples and sequencing controls, the extent to which one can do so is limited by the stochastic nature of contaminant introduction during the sequencing workflow. Given the compositional nature of microbiome data, it would not be correct to assume that the reads detected in the negative controls represent the absolute limit of contamination introduced into the dataset. Accordingly, one could not conclude that all the reads observed in a meconium sample, but absent from its paired control, were truly present in the foetal gut. However, we have now conducted an additional analysis using the ALDEx2 package in R, demonstrating that there are no differentially abundant taxa between the meconium samples and sequencing negative controls. Supplementary File S1 now contains the output of ALDEx2, along with an explanation of the column headers and a description of the code to replicate the analysis in R. We have edited the methods (lines 202-204; new manuscript), results (lines 265-267; new manuscript) and discussion (lines 356-357; new manuscript) accordingly:

Methods: "The aldex wrapper function was used to assess differential abundance between microbiome profiles of meconium and negative controls."

Results: "In line with this, no taxa were found to be differentially abundant in meconium samples compared to sequencing negative controls using the ALDEx2 package ($P > 0.831$ for all taxa)."

Discussion: "This is underscored by the absence of any differentially abundant taxa identified by ALDEx2 (Supplementary File S1)."

Regarding the read numbers of the controls used in this study, there does indeed appear to be a difference between the number of reads among DNA extraction controls included in this study. Since submitting this manuscript for peer review, we have completed sequencing of all samples collected for the Maternal Microbiome Legacy Project, including 212 sequencing controls. Although the majority of these 212 controls were not generated during the DNA

extractions and PCRs relevant to this study, the distribution of read numbers among the entire control dataset span from <10, to approximately 2,000 reads, with the bulk of the read numbers containing <100 reads (see graph below; not for publication). The read numbers of the 19 controls are in fact quite representative of the overall dataset.

2. The authors must provide quantifications of β -diversity to compute community dissimilarity scores instead of directly plotting the read-numbers. Computing Bray-Curtis Dissimilarity, Weighted UniFrac distance or any other similar approach would be an accurate demonstration of dissimilarity among case and controls.

Authors should also give an assessment of alpha-diversity by comparing the taxonomic dissimilarity among the meconium samples themselves, to address the possibility of common contaminants originating from reagents, Hospital contaminations etc as mentioned by the authors.

Response: We thank the reviewer for the above comment about calculating alpha and beta-diversity metrics which are frequently calculated for microbiome studies using pipelines such as QIIME2. In this case, calculating these metrics would not yield useful information as this requires rarefaction of sequencing reads to a specified depth. Given that the vast majority of the meconium samples described in this study have very low read numbers, rarefaction to an appropriate number of reads would exclude almost all of the samples. Accordingly, we opted to present the most common taxa identified across all samples and composition of the negative controls in figures (Figures 1B and 1C, respectively) and describe the number of samples comprising a single NN and the number of samples dominated by a single NN rather than the above metrics.

Additionally, we opted to analyse our data using compositional methods, rather than calculating Bray-Curtis dissimilarity or UniFrac, which are not an appropriate choice for compositional data (see Gloor et al. 2017. *Front Microbiol*; **8**: 2224). Instead, we performed centre log-ratio transformation on the feature table of nearest neighbours using the ALDEx2 package and employed principal components analysis on the resulting matrix (Figure 2), a suitable compositional method analogous to beta diversity exploration using BC/UniFrac.

3. The authors should discuss the bacterial genera identified in their analysis especially for the samples above 1000 read-counts and compare them with the genera identified in infant stool samples. If the apparent increase in microbial reads is indeed due to the time spent post birth, it would be interesting to see if they match the profile of an infant stool.

Response: We thank the reviewer for pointing out this omission. Accordingly, we have made a new supplementary figure (Supplementary Figure S4) showing microbiome compositions and total read numbers for these 20 higher read count samples, as recommended by the reviewer, and confirmed that these compositions are similar to that of infant stool (lines 257-259; new manuscript):

“The microbial composition of this subset of meconium samples, closely matched that of stool from 3-month-old infants also enrolled in the LEGACY study (Dos Santos et al., unpublished data).”.

While we have mentioned the comparable read numbers of the infant stool samples and the 20 meconium samples with higher read numbers, we hope the reviewer understands our reluctance to discuss their composition in any great detail. First and foremost, we are already pushing the word limit constraints of the journal, but also, we are currently analysing and preparing a manuscript regarding the entire set of infant stool microbiomes collected as part of the LEGACY study. These infant stool microbiomes are also part of an abstract submission to conferences which prohibit prior publication of study data, even in part. We can however confirm that the reviewer is correct: the composition of the 20 high-read meconium microbiomes matches that of the infant stool samples collected 3 months postpartum.

4. Did the authors normalize the weight of meconium and stool for each sample for qPCR? What was the exact weight taken for the qPCR of 16S rRNA based microbial profiling?

Response: No normalisation as described by the reviewer was necessary. All DNA extractions were performed on 200 mg of meconium, exactly. The same DNA extract (2 μ L) was used for both the *cpn60* microbiome profiling and the qPCR of the 16S rRNA gene.

5. In the culture experiments, it is not clear if the samples were inoculated freshly or post cryo-storage? If the samples are not freshly inoculated, most of the bacterial taxa won't survive post cryo-storage and what authors may observe is a selective enrichment of certain bacterial species that may survive cryo-preservation or have been introduced during culturing. Additionally, were there appropriate PBS controls taken during inoculation? If yes, their culture profile should also be discussed.

Response: We thank the reviewer for raising this important point, which we can certainly clarify. After collection in Vancouver, BC, diapers containing meconium were sent to the clinical processing lab for aliquoting into cryovials and storage at -80°C until they could be shipped to the sequencing lab in Saskatoon, Saskatchewan. The samples in the present study were all collected from March to September 2018, and culture work was undertaken from August to November 2019, meaning samples were stored on average for just over a year.

We agree that there is certainly some effect of cryopreservation on the viability of bacteria within these meconium samples, although long-term storage of samples at -80°C in large-scale studies like ours is common. However, the aim of our culture experiment was not to provide an exhaustive list of species that can be cultured from meconium, but rather to determine if there were viable bacteria in the sample at all. However, the reviewer's comment is important, and we have now explicitly stated the goal of our culture experiment (lines 405-406; new manuscript), and acknowledged the limitation posed by the length of cryopreservation (lines 406-407; new manuscript):

Discussion: “While the use of a single set of culture conditions can be considered a limitation in our study, the purpose of attempting cultivation was not to compile an exhaustive list of species present in meconium, but rather to determine the presence of any viable organisms.”

Discussion: “Likewise, the long-term storage (~1 year) of samples at -80°C prior to culture may also have reduced bacterial viability.”

Regarding selective enrichment of the isolated taxa, the reviewer is quite correct to point this out. The main three genera we isolated are indeed very well adapted to growth *in vitro* and we mention this fact in the original manuscript (lines 371-373; original manuscript). Finally, with respect to PBS controls, we did indeed plate out 100 µL of the PBS used for homogenisation for each batch of meconium samples cultured. No growth was observed for any of these controls. We have updated the methods and results sections to reflect this (line 181 and lines 304-306; new manuscript). Indeed, if any contaminants were present in the PBS, they would be easy to detect as a similar number of colonies of the respective contaminant would be present on all plates, regardless of dilution.

Methods: “PBS used for suspension of meconium was also plated to check for contaminants.”.

Results: “No growth was detected on PBS control plates, while culture-positive plates inoculated with undiluted meconium (see methods) only grew a few colonies under aerobic conditions (Supplementary Figure S6)”.

6. Authors have only provided aerobic culture environment to the isolates, which is suitable for the infant stool samples but not for culturing the microbes from meconium samples which would be of fetal origin. The oxygen supply to the fetal compartment during gestation is quite limited and changes through gestational timeline. Selective enrichment in aerobic cultures will result in the loss of anaerobic microbes likely present in fetal meconium.

Response: The reviewer is quite correct to point out this limitation of our study, which we have also discussed in the original manuscript (lines 381-384). Indeed, we realise that additional species would likely have been detected if all samples underwent anaerobic culture. However, as outlined in the response to comment 5, we were not attempting to compile a complete collection of isolates that could be cultured from meconium, but rather to determine if any viable organisms were present in the sample.

Given the culture-independent nature of microbiome profiling, we did indeed detect enteric anaerobes in some meconium samples; however, the samples containing more than one hundred reads of *Bacteroides* / *Parabacteroides*, for example, were also the high read number samples which were thought to be stool rather than meconium (see also Supplementary File S1). Aside from these samples, very few reads of these genera were detected across the microbiome dataset. As a result, although we cannot comment on the culture of anaerobic species from meconium, their presence as a key part of the meconium microbiome would have been detected by the *cpn60* amplicon sequencing.

7. How do authors explain 'No significant difference in isolate distribution was observed when grouping by read count' in their culture conditions? If the samples with >1000 READS are indeed due to meconium samples being of later time points (as explained by the meconium appearance) or are contaminated, why they did not yield significantly high viable microbes in culture?

Response: We thank the reviewer for this important comment and offer the following explanations. The media we used for culturing meconium is selective, and so even if there were a higher number of reads, it may be the case that the dominant taxa's growth requirements were not met by this medium. Similarly, as discussed in the response to comment 6, six of the high read number samples were dominated by obligate anaerobic species which would not have grown on the medium. Finally, as the reviewer points out in comment 5, the cryopreservation duration may also have impacted bacterial viability, and this is now acknowledged in the discussion section (see response to comment 5).

8. Authors must also discuss in their study some of the previous reports where human meconium has been associated with detectable microbial presence. Some of them are mentioned below. Also they should discuss some new reports on the presence of microbes in human fetal gut.

Stinson, Lisa F., et al. "The not-so-sterile womb: evidence that the human fetus is exposed to bacteria prior to birth." *Frontiers in microbiology* 10 (2019): 1124.

He, Qiuwen, et al. "The meconium microbiota shares more features with the amniotic fluid microbiota than the maternal fecal and vaginal microbiota." *Gut Microbes* 12.1 (2020): 1794266

Ardissone, Alexandria N., et al. "Meconium microbiome analysis identifies bacteria correlated with premature birth." *PloS one* 9.3 (2014): e90784.

Gosalbes, M. J., et al. "Meconium microbiota types dominated by lactic acid or enteric bacteria are differentially associated with maternal eczema and respiratory problems in infants." *Clinical & Experimental Allergy* 43.2 (2013): 198-211

Hu, Jianzhong, et al. "Diversified microbiota of meconium is affected by maternal diabetes status." *PloS one* 8.11 (2013): e78257

Response: We thank the reviewer for this selection of literature describing the detection of various microbes in meconium. In the original manuscript, we briefly summarised the various bacterial taxa detected in other sequencing studies of meconium (lines 338-341). We have now opted to cite several of the papers mentioned above in our revised manuscript in accordance with the reviewer's recommendation as they illustrate well the problem of contamination in microbiome studies of samples with inherently low microbial biomass.

The studies by Hu *et al.*, Gosalbes *et al.* and Ardissone *et al.* either did not include any controls (Hu / Gosalbes) or included DNA extraction controls but relied on gel electrophoresis after 16S rRNA PCR to assess "sterility" and lack of contamination (Ardissone). This approach is not valid for determining if a sample is affected by contamination, as a sample or control showing no band after PCR and gel electrophoresis can still contain thousands of reads. In effect, one *must* sequence the negative controls to gain any insight into the extent of contamination. The recent study by He *et al.* suffers from the same problem, which is also acknowledged by the authors in their discussion:

"Another significant limitation was that contaminant and extraction blank controls were not included; thus, this work could not completely rule out the chance of contamination acquired during the experimental workflow."

Additionally, the principle finding of this paper is that the microbial content of meconium is far more similar to that of amniotic fluid than that of the maternal vaginal or gut. Given that

amniotic fluid is also a low microbial biomass environment, and that there have been numerous recent, properly controlled studies showing the prevalence of contamination in amniotic fluid microbiomes, this is further evidence that meconium (along with the placenta and amniotic fluid) does not contain a unique, functional microbiome (see original manuscript for supporting references).

Finally, the paper by Stinson *et al.* is an excellent case study of why negative controls are required- not optional- in microbiome studies of low microbial biomass environments. The authors describe a set of 43 meconium samples, of which, “[thirty-eight] samples contained high numbers of reads that mapped to *Pelomonas puraquae* with a 99.5% sequence homology”. Figure 1 of this paper shows that both *P. puraquae* and its sister species, *P. aquatica* completely dominate the meconium microbiome profiles, with tens of thousands of reads in all but five samples. This species has previously been identified as a contaminant in studies of commercial DNA extraction kits and ultrapure water (see Results of Stinson *et al.* for references) and have also been associated with environmental pollutants in studies of inflow and pond water from trout farms (Mahmood and Magdy, 2021. *Sci Rep*; **11**: 421). Moreover, these species were detected in the extraction controls of the same study by Stinson *et al.* As a result, one cannot reasonably conclude these species are truly present within the foetal gut. They also noted the detection of several taxa in meconium samples which “are not biologically plausible human microbiome candidates”, representing thermophilic species and likely contaminants.

Accordingly, we have now cited these studies in the introduction and discussion of the revised manuscript:

Introduction: lines 85 and 90.

Discussion: line 342 and lines 343 – 347, “When controls are included, contamination becomes evident: one meconium study detected thousands of reads aligning to *Pelomonas puraquae*, a species isolated from industrial water and associated with environmental pollutants in marine studies(52,53), in 38/43 samples and within negative controls. Previous work has identified *Pelomonas* spp. as contaminants of extraction kits and ultrapure water(20,22).”

August 26, 2021

Dr. Deborah M. Money
University of British Columbia
Department of Obstetrics and Gynaecology
Vancouver, BC
Canada

Re: Spectrum00067-21R1 (Early neonatal meconium does not have a demonstrable microbiota determined through use of robust negative controls with *cpn60*-based microbiome profiling.)

Dear Dr. Deborah M. Money:

Your manuscript has been accepted, and I am forwarding it to the ASM Journals Department for publication. You will be notified when your proofs are ready to be viewed.

Sincerely,

Zhenjiang Xu
Editor, Microbiology Spectrum

Journals Department
Supplemental Dataset: Accept
Supplemental Material: Accept